# Improved Sampling in Ab Initio Free Energy Calculations of Biomolecules at Solid–Liquid Interfaces: Tight-Binding Assessment of Charged Amino Acids on TiO$_2$ Anatase (101)

**Lorenzo Agosta** [1,*] **, Erik G. Brandt** [2] **and Alexander Lyubartsev** [2]

1   Department of Chemistry, Ångström Laboratory, Uppsala University, 75237 Uppsala, Sweden
2   Department of Materials and Environmental Chemistry, Stockholm University, S-10691 Stockholm, Sweden; erik.brandt@mmk.su.se (E.G.B.); alexander.lyubartsev@mmk.su.se (A.L.)
*   Correspondence: lorenzo.agosta@kemi.uu.se

**Abstract:** Atomistic simulations can complement the scarce experimental data on free energies of molecules at bio-inorganic interfaces. In molecular simulations, adsorption free energy landscapes are efficiently explored with advanced sampling methods, but classical dynamics is unable to capture charge transfer and polarization at the solid–liquid interface. Ab initio simulations do not suffer from this flaw, but only at the expense of an overwhelming computational cost. Here, we introduce a protocol for adsorption free energy calculations that improves sampling on the timescales relevant to ab initio simulations. As a case study, we calculate adsorption free energies of the charged amino acids Lysine and Aspartate on the fully hydrated anatase (101) TiO$_2$ surface using tight-binding forces. We find that the first-principle description of the system significantly contributes to the adsorption free energies, which is overlooked by calculations with previous methods.

**Keywords:** free energy; metadynamics; adsorption; TiO$_2$; amino acids; ab initio; Tight-Binding

## 1. Introduction

Engineered inorganic materials are important in many technological applications [1–3] such as biomimetics [4], optics [5], biosensors [6,7], and smart surfaces [8]. The key to harness the true potentials of these and other nanobio applications lies in a microscopic understanding of biomolecular adsorption on inorganic materials. Nanotoxicity is another area where molecular interactions determine the effects of nanoparticles on biological organisms, and controls the outcome in terms of nanomaterial safety [9,10]. The bionano region, the nanometer-thick boundary near the surfaces of nanoparticles and engineered nanomaterials, is believed to regulate adsorption behavior [9,11]. This region can be probed with modern atomistic simulations, but is difficult to access in experiments due to the weak signal generated by its comparatively small volume.

Titanium dioxide (TiO$_2$) is a biocompatible semiconductor used in implants and biomedical applications [12–14] with a low bandgap that is ideal for water-splitting applications [15]. TiO$_2$ has become the standard surface model for water interactions with metal oxides in theoretical chemistry due its perceived simplicity, with reactive sites at undercoordinated titanium atoms (Ti$_{nc}$) and bridging oxygen atoms (O$_{br}$) [16,17]. However, on this "simple" surface, the adsorption behavior is modified by surface hydroxyl groups from spontaneous water splitting [18–20], and adsorbate interactions are indirectly mediated via strongly adsorbed water layers [21–23]. Charge transfer and polarization at the surface are other important factors to consider [24].

The binding free energy of a biomolecule on nano-TiO$_2$, $\Delta F_{\text{ads}}$, dictates the adsorption behavior in equilibrium. This quantity can be measured experimentally [25] and calculated from simulations, but the latter demands careful evaluation of all energy and entropy contributions at the solid–liquid interface. The Lennard–Jones models with fixed partial Coulomb charges employed in classical molecular dynamics (CMD) simulations may not be sensitive to subtle differences in adsorption behavior between similar biomolecules on, e.g., TiO$_2$. Certainly, much effort has been devoted with such force fields to compute trends for amino acids on various TiO$_2$ surfaces [26–29], amorphous nanoparticles [30], and other inorganic materials [31]. The overall assessment is that amino acids bind strongly if they can penetrate the first water layer at the solid–liquid TiO$_2$ interface [32], but many questions remain unanswered. For example, the force fields used to model TiO$_2$–adsorbate interactions are parameterized on small and neutral molecules, where polarization and charge transfer take a small role. Further, there are significant differences in the water density profiles near TiO$_2$ obtained with density functional theory (DFT) [32] and CMD, respectively.

Overestimated surface/water interactions lead to artificial order at the interface, which prevents direct molecular adsorption [29]. Underestimation, on the other hand, smooths out the distinctive adsorption behavior of individual biomolecules [26–28]. These differences in force field parameterizations sometimes result in different adsorption behavior for the same molecules [26–29]. In the absence of experimental evidence, ab initio simulations that account for reactivity and polarization at the solid–liquid interface can resolve these differences. Ab initio molecular dynamics simulations (AIMD), however, are limited to tens of ps of simulation time, which renders proper sampling impossible.

Here, we validate a simulation protocol that substantially improves sampling in free energy calculations with ab initio dynamics. This method can be used to calculate accurate adsorption free energies of small (even charged) molecules on inorganic (and other) surfaces. This approach is based on semi-empirical Tight-Binding (TB) forces, which captures reactivity, polarization, and charge transfer [33], and can be used for orders-of-magnitude longer simulations compared to DFT with the Generalized-Gradient Approximation (GGA) for the forces [34,35]. We use multi-walker metadynamics [36] with an augmented collective variable (CV) that significantly improves sampling of the target CV compared to standard metadynamics. Furthermore, we reconstruct free energies using a "mean force estimator", which is superior to the traditional way of cumulatively summing the bias potentials [37].

We show that this combination of advanced simulation techniques enables the calculation of ab initio-based, converged, free energy profiles for small molecules on inorganic surfaces. As a case study, we calculate the adsorption free energies of the charged amino acids Lysine (Lys) and Aspartate (Asp) on fully hydrated TiO$_2$ anatase (101) surfaces. These molecules have been shown to interact strongly with TiO$_2$ surfaces in previous single-point DFT calculations, with adsorption energies that failed to be explained with classical models [23,33,38] .

## 2. Methods

### 2.1. System Preparations

The ab initio molecular dynamics (AIMD) simulations using self-consistent density functional tight-binding (DFTB) approach [39] were done with Cp2K [40] with the same setup as in [32]. Briefly, we used PACKMOL [41] to prepare anatase (101) simulation boxes with sizes $10.35 \times 11.4 \times 43$ Å. The TiO$_2$ slabs were built by repeating the unit cell four times along the z-direction (same as in our previous simulations [32]), and the rest of the boxes were filled with one amino acid and water amounts that correspond to 1 atm and 310 K [32]. These simulation boxes are large enough to avoid self-interactions. We used analog molecules for Aspartate and Lysine, i.e., side chains of the amino acids with the backbone replaced with a CH$_3$-group. The removal of the backbone mimics the state of the amino acid in a protein, where the backbone is buried and prevented to contribute to the adsorption

except at the terminal groups. We kept the systems neutral with a counterion ($OH^-$ or $OH_3^+$) initially placed on the opposite surface slab to the amino acid.

Tight-binding (TB) calculations with amino acids have been reported on dry TiO$_2$ surfaces [33] but, to the best of the authors' knowledge, not at full hydration. We therefore first tested the Matsci [34] and Mio-Tiorg [35] TB parameterizations for lysine and aspartate at TiO$_2$–water interfaces. In both cases, we found that the amino acid $C - H$- and $N - H$-groups deprotonated on the TiO$_2$ anatase (101) surface. There is no physical basis for such behavior, which implies that the overlap integrals of the underlying interactions are underestimated. We added harmonic constraints on the $C - H$- and $N - H$-groups as a simple remedy in the rest of the simulations. Further, the test simulations revealed geometrical distortions at the solid–liquid interface with Mio-Tiorg parameters, but structures were in good agreement to DFT-derived geometries with the Matsci parameters [32]. Based on these results, we used the Matsci parameterization, without long-range dispersions, for all free energy calculations in this work. Furthermore, the final adsorbing modes of Lys and Asp were fully optimized both with TB and DFT approaches [32] to check the consistency using two different levels of theory. For the DFT calculations, we used the BLYP functional [42,43] augmented with the Grimme DFT-D3 dispersion corrections [44], whereas the GTH normconserving pseudopotentials [45,46] and a double-$\zeta$ Gaussian basis set with polarization functions (DZVP) [47] were used to describe the core and valence electrons respectively. The energy cut-off for the electron density expansion in the GPW method was 400 Ry and the minimization was stopped when the total forces were lower than $10^{-3}$ Hartree/Bohr.

### 2.2. Metadynamics Setup

Metadynamics simulations were carried out with the PLUMED [48] module in CP2K. Specifically, we used well-tempered metadynamics with adaptive Gaussians (AWTMetaD) [49–51] and a bias factor of 15. We started with Gaussian heights of $3.5\,kJ\,mol^{-1}$, added new bias potentials every 25 fs, and adjusted the widths of the Gaussian every 75 fs. This parameter combination is a reasonable compromise of accuracy vs. speed for ab initio dynamics with limited time sampling [52].

We used the surface separation distance (SSD) as the target collective variable (CV) in the free energy calculations. The SSD is the distance between the outermost layer of Ti surface atoms and the center of mass of a group of concern in the amino acid, $NH_3^+$- and $COO^-$-groups respectively for Lys and Asp. In addition, we augmented the calculations with a second collective variable with the aim of boosting the exploration of the adsorption landscape along another important degree of freedom—the adsorbent orientation—thus improving the sampling of the target CV [53]. As augmenting variable, we used the angle between the vector of the surface normal and $N-C_{fn}$ (in Lys) or $C_{OO^-}-C_{fn}$ (in Asp), where $C_{fn}$ is the first carbon neighbor to the atom in question . The augmented variable is integrated out during the calculation of the free energy (see Equation 2). We restricted the phase space with a wall potential 1 nm away from the outermost Ti atoms, and launched multi-walker [36] AWTMetaD with eight replicas starting at different CV values. The walkers communicated every 25 fs and we simulated 240–300 ps for each walker.

Traditionally, the potential of mean force (PMFs) is reconstructed from the accumulated bias potentials [29,49,51]

$$W_2(z, \theta) = -\lim_{t \to \infty} \left( V(z, \theta, t) + k_B T \ln \left( n(z, \theta, t) \right) + const \right) \tag{1}$$

where $V(z, \theta, t)$ is the accumulated bias potential, $\theta$ is the augmenting CV, and $n(z, \theta, t)$ is the accumulated histogram of the target collective variable. This correction term from the histogram is needed when the widths of the Gaussians bias are adjusted dynamically [51]. Integrating out the augmenting collective variable $\theta$ yields to the potential of mean force along the main collective variable [53]:

$$PMF(z) = -k_B T \ln \int e^{-W_2(z, \theta)/k_B T} d\theta + const \tag{2}$$

Forces can be used directly in the calculations to improve convergence in free energy profiles from metadynamics [37]. With this in mind, we also calculated free energy profiles by thermodynamic integration of $F(z)$, the mean force on the $NH_3^+$- and $COO^-$-groups in the metadynamics simulations (more exactly, from the derivative of the energy over the SSD collective variable, not including the bias energy). In this formulation the PMF is calculated as

$$\text{PMF}_{\text{TI}}(z) = -\int_{r_c}^{z} \langle F(z') \rangle \, dz' + \text{const}, \tag{3}$$

where $z$ spans the SSD-values from $r_c$ (the onset of the solid surface) to 1 nm, where the potential wall is set. Canonical averaging of the average force is evaluated from all sampled configurations "$i$" having the main collective variable $z$ in the range $z - \Delta z/2 < z_i < z + \Delta z/2$:

$$\langle F(z) \rangle = \frac{\sum_i F(i) exp\left(-V(z; \theta_i, t)/k_B T\right)}{\sum_i exp\left(-V(z; \theta_i, t)/k_B T\right)} \tag{4}$$

In our computations, bin size for force integration was set to $\Delta z = 0.005$ Å. We will refer to this method as MetaDF (metadynamics with force integration) in the rest of this text. Note that MetaDF is not bound to a specific implementation of the metadynamics. It can use well-tempered metadynamics, or constant Gaussian height metadynamics, as it follows from Equations (3) and (4).

The binding free energy is computed from the PMF as [29]

$$\Delta F_{\text{ads}} = -k_B T \ln\left(\frac{1}{\delta} \int_{r_c}^{r_c+\delta} e^{-\text{PMF}(z)/k_B T} dz\right), \tag{5}$$

where $k_B T$ is the product of the Boltzmann constant and the absolute temperature, $\delta = 8$ Å is the thickness of the adsorption layer and $r_c + \delta$ is the start of the liquid bulk. Equation (5) is the thermodynamically correct route to $\Delta F_{\text{ads}}$, but earlier work has also quantified the adsorbate's binding strength by the difference of the minimum and bulk values of its PMF [26–28]

$$\Delta F_{\text{diff}} = \text{PMF}(r_c + \delta) - \text{PMF}(r_c) \tag{6}$$

This difference is always larger than $\Delta F_{\text{ads}}$. Note that although $\Delta F_{\text{diff}}$ depends on the specific choice of the CV (e.g., determined by the molecular center of mass or by a specific atomic group on a sorbent molecule), $\Delta F_{\text{ads}}$ does not depend on such choice.

## 3. Results and Discussion

### 3.1. Method Validation

To validate the new protocol described so far and the parameters chosen for ab initio Metadynamics, we run extended classical molecular dynamics (CMD) simulations for the lysine–anatase (101) system. The force field describing the interactions was taken from reference [29]. We run classical simulations with the same system size and the same MetaD parameters as we used in ab initio TB computations, but we extended classical simulations up to 200 ns using only the SSD variable for sampling the free energy and comparing the effect of having a single or eight walkers. We tested the effect of the Gaussian height and insertion rate on the PMF convergence. Eight different Metadynamics simulations were run for the single walker simulations with different combinations of these parameters. Gaussians heights and insertion rate were spanning 1 to 3.5 kJ/mol and 25 to 500 fs respectively. In each case for 200 ns of classical simulations no significant difference in the PMF profiles obtained at different Gaussian heights and insertion rates were noticed.

In Figure 1, the convergence test is shown for MetaDF compared to AWTMetaD for Gaussians height = 3.5 kJ/mol and insertion rate = 50 fs. Two adsorption modes can be distinguished: one $M_{med}$

where the Lysine-TiO$_2$ interaction is mediated by water at (SSD = 5 Å), and a second one $M_{dir}$ where a direct surface contact occurs on the O$_{br}$ at SSD = 3.2 Å. Both MetaDF and AWTMetaD converge to the same PMF profile after 200 ns as expected. The main difference is that although MetaDF needs barely 1 ns to reach the final profile on the $M_{med}$ mode, AWTMetaD requires 30 ns. This effect is even more pronounced for the second adsorption mode $M_{dir}$ where the AWTMetaD convergence is strongly hindered by the long time sampling of the transition between direct-mediated adsorption modes and the cumulative behavior of the bias potential which changes strongly if the CV falls in a previously poorly sampled region. On the contrary, MetaDF converges once the main regions of the free energy landscape have been scanned due to the fact that the average force acting on the CV does not change significantly with long sampling. When eight walkers are implemented (Figure 1b, bottom) the final PMF profile for the $M_{med}$ adsorption mode is reached after barely 300 ps with MetaDF due to the fact that each walker push the remaining walkers to explore other CV values, thus the whole range of CV is sampled simultaneously. The second mode $M_{dir}$ is also sampled faster than with a single walker and results, after 300 ps, in a profile rather close to that obtained in 200 ns calculations. This set-up shows that the convergence can be obtained at least 100 times faster than using a single walker with AWTMetaD. Finally, we note that the adsorption free energy value of Lysine arising uniquely from the $M_{med}$ mode is $F_{ads} = -2.7 \, \text{kJ mol}^{-1}$, indicating that in the classical description Lysine adsorbs very weakly on the anatase (101) surface.

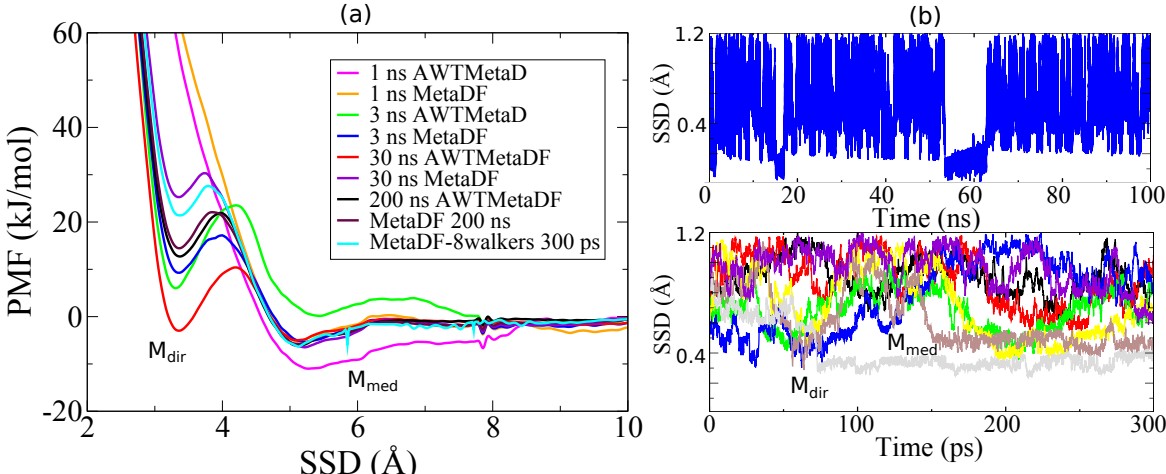

**Figure 1.** (**a**) Potentials of mean force (PMF) from CMD along the surface separation distance (SSD) for Lysine on anatase (101) calculated with AWTMetaD and MetaDF at different intervals of time. (**b**) Single (up) and 8 walkers (down) dynamics used to test the convergence. The two minima in the PMF correspond to a water mediated adsorption mode $M_{med}$ (SSD = 5 Å) and a direct surface contact mode $M_{dir}$ on the O$_{br}$ at SSD = 3.2 Å. AWTMetaD and MetaDF converge using a single walker in the long time range (200 ns) to the same PMF profile. The mode $M_{med}$ converges within 1 ns with an error of 0.6 kJ mol$^{-1}$ with MetaDF while AWTMetaD converges after 30 ns within the same error. Adopting 8 walkers permits to have the same convergence in only 300 ps with MetaDF and to sample the $M_{dir}$ mode about 100 time faster compared to AWTMetaD.

### 3.2. Tight-Binding Results

Charged amino acids are driving adsorption at bio-inorganic interfaces [23,38,54,55], and are the most challenging to model since they can induce strong polarization at interfaces and change the surface's protonation state. In this work, we used Lysine and Aspartate (+1 and −1 charge, respectively) to investigate how charged amino acids impact the adsorption free energy in bionano systems.

Figure 2c shows the eight walkers exploring the SSD during the Lysine TB simulation. By analyzing contact configurations, we identified two adsorption modes, corresponding to double ($L_I$)

and single ($L_{II}$) adsorption on oxygen bridges ($O_{br}$) on the TiO$_2$ surface) (Figure 2a,b). The real strength of the multi-walker method is not in the brute force sampling itself, but how walkers in different regions of the free energy landscape communicate so that, although an individual walker may sample a limited area of the configurational space, all walkers together sample the whole relevant region. This effect is extremely important if combined with the force estimator for the calculation of the PMF, as explained in the validation section for CMD. Multi-walker metadynamics also yields linear scaling to reconstruct the free energy landscape with required precision, whereas single-walker metadynamics is limited by slow diffusion [36]. In the case of adsorption at the TiO$_2$–water interface, individual walkers penetrate into the elusive adsorbed water layer adjacent to the surface. This region is extremely difficult to sample under normal circumstances [56] (which is also illustrated by our CMD simulations described in the previous section), but multiple walkers solve this problem. Figure 2c shows that several adsorption/desorption events are sampled by different walkers during the calculation, as necessary to obtain proper equilibrium statistics. Figure 2d shows the cumulative SSD distribution from all walkers. The peak in the SSD histogram associated to adsorption modes $L_I$ and $L_{II}$ is significantly pronounced after 300 ps of simulation per walker, and is still accentuated when calculated on the last 50 ps of the simulation trajectory. A flat profile in the CV histogram is the signature of a converged free energy profile when calculated with the standard estimator (Equation (2)). This emphasizes how slow PMF convergence can be in standard metadynamics compared to integrating forces (Equation (3)), which does not depend on the accumulated histogram.

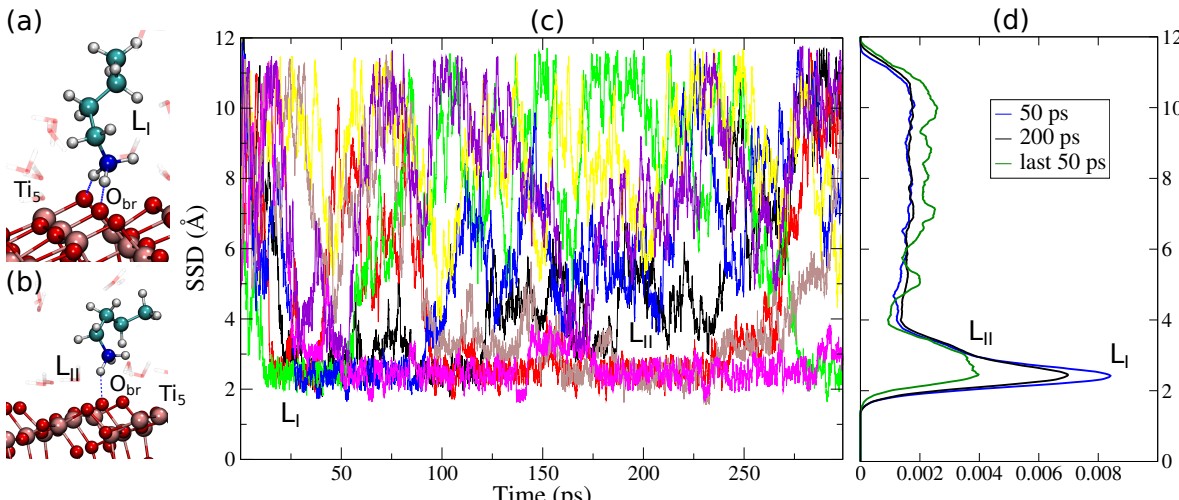

**Figure 2.** (**a**,**b**) Snapshots of adsorption modes $L_I$ at SSD = 2.5 Å and $L_{II}$ at SSD = 3.3 Å for Lysine on anatase (101). The freely diffusive water layers begin at SSD = 4 Å. (**c**) The surface separation distance (SSD) for the eight walkers used in the free energy calculations of Lysine, showing that the target collective variable is adequately sampled in 50 ps per walker. (**d**) The cumulative SSD distribution from all walkers at different times. No significant difference in the histogram is found after 50 ps per walker. The last 50 ps indicates that convergence is not reached with AWTMetaD after the full simulation time.

Figure 3 shows potentials of mean force (PMFs) along the SSD for Lysine on anatase (101) calculated with MetaDF and AWTMetaD. $\theta$ is also completely sampled (see Figure S1 in the Supporting Information for the 2D map), and it shows a single global minima at the adsorption site. The PMFs are plotted at different times up to 300 ps per walker and show the convergence behaviors of the two methods. The PMFs calculated with the standard estimator (Equation (2)) fluctuate strongly, which is a manifestation of slow convergence after 300 ps of simulated time (Figures 2d and 3d). The PMFs calculated by integrating forces on the NH$_3{}^+$-group converges to a smooth profile with an error of 3 kJ mol$^{-1}$ (as estimated from the force variance [57]) after 250 ps per walker. This is visible also from Figure 3d, where $\Delta F_{ads}$ is plotted as a function of time for MetaDF and AWTMetaD. Via Equation (5), we calculate that the binding free energy for Lysine on anatase (101) is $\Delta F_{ads} = -58.2$ kJ mol$^{-1}$

(or $\Delta F_{\text{diff}} = -65.4\,\text{kJ}\,\text{mol}^{-1}$ from Equation (6)). This is substantially larger (3–10 times) than values reported from different CMD simulations [26–29], where both direct and indirect adsorption modes were found. This difference suggests that electronic degrees of freedom, in particular polarization and charge transfer effects at the interface, cannot be neglected in the free energy calculations.

To further validate the TB parametrization used in our calculations, we performed optimization computations for the $L_I$ mode bound state using TB and DFT-GGA theory. The optimized bond length between the hydrogen of the amino group and the bridging oxygen ($\text{NH}-\text{O}_{\text{br}}$, Figure 2a) was found to be $d_{\text{NH}-\text{O}_{\text{br}}} = 1.7\,\text{Å}$ for DFT-GGA and $d_{\text{NH}-\text{O}_{\text{br}}} = 1.6\,\text{Å}$ for TB. No further discrepancies or adjustment in the final adsorption configuration were noticed.

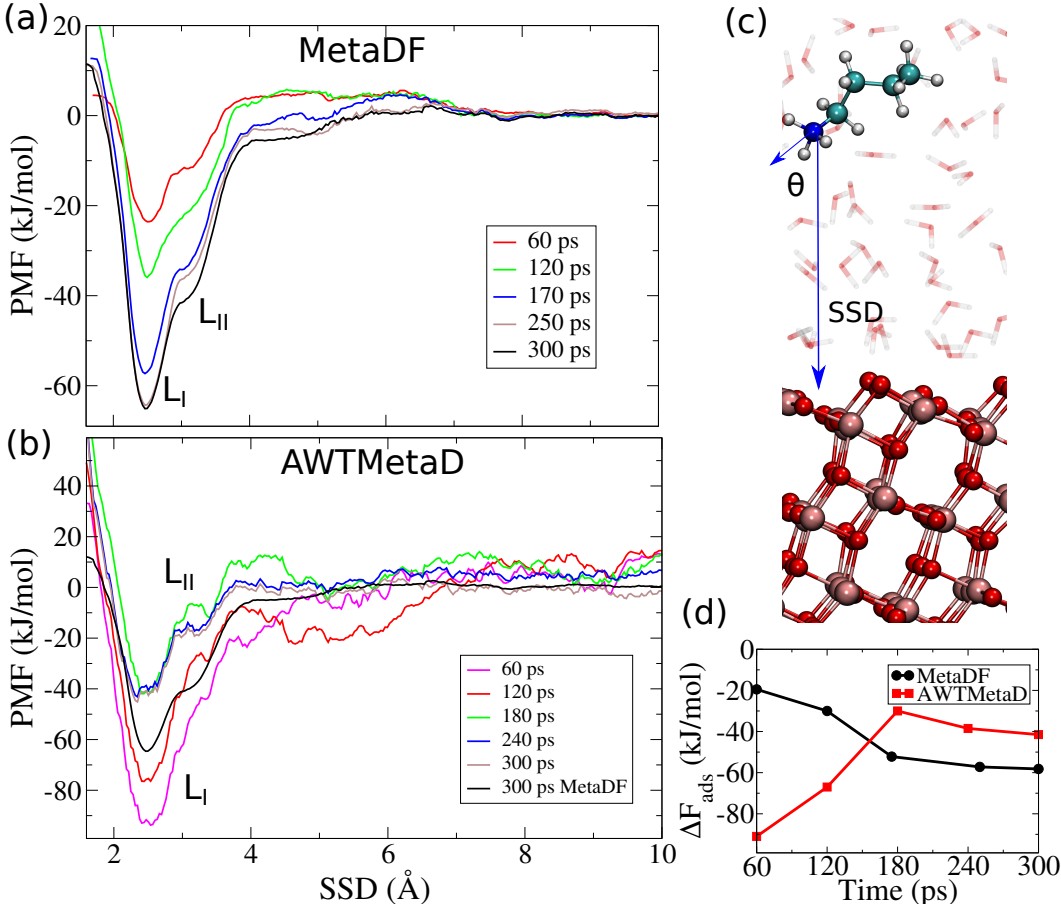

**Figure 3.** Potentials of mean force (PMFs) for Lysine on anatase (101) along the surface separation distance (SSD), calculated with (**a**) MetaDF and (**b**) AWTMetaD. The PMF converges to an error of $3\,\text{kJ}\,\text{mol}^{-1}$ (as estimated from the force variance [57] and the convergence profile (**d**)) within 250 ps of MetaDF, but fails to converge after 300 ps of AWTMetaD. The reported times are per walker, with eight walkers per calculation. (**c**) A representation of the two CVs used to sample the free energy landscape. $\theta$ is the angle between the normal vector to the TiO$_2$ surface and the vector $\text{N}-\text{C}_{\text{fn}}$. (**d**) $\Delta F_{\text{ads}}$ as function of time. MetaDF converges asymptotically after 250 ps but AWTMetaD does not converge within 300 ps.

For Aspartate on anatase (101), we also found two adsorption modes: $A_I$, which is not in direct contact with the surface but mediated via the first water layer, and $A_{II}$, which is in direct contact with the TiO$_2$ surface (Figure 4a,b). Figure 5 shows the difference in PMFs calculated with MetaDF compared to the standard free energy estimator. Mode $A_I$ is well-sampled in 180 ps with MetaDF (within an error of $3\,\text{kJ}\,\text{mol}^{-1}$ as estimated from the force variance [57] and inspecting $F_{\text{ads}}$ as a function of time as shown in Figure 5d), while AWTMetaD (Equation (2)) fails to converge after 300 ps of simulation time (Figures 4d and 5d). The $A_{II}$-mode (direct contact) appears when a single walker penetrates the

first surface water layer after 60 ps. The $A_{II}$-mode is separated from the $A_I$-mode by a free energy barrier of $\sim 20\,\mathrm{kJ\,mol^{-1}}$, and thus much more challenging to sample. The high barrier and narrow region available to $A_{II}$ implies that the main contribution to the binding free energy is coming from $A_I$. The binding free energy is $\Delta F_{\mathrm{ads}} = 12.1\,\mathrm{kJ\,mol^{-1}}$ from Equation (5) (or $17.2\,\mathrm{kJ\,mol^{-1}}$ via Equation (6)). CMD simulations have reported similar values of the binding free energy when the $\mathrm{COO^-}$-group is in direct contact to the surface [58] and via surface hydroxyls [59]. A water-mediated adsorption mode has not been found due to the weak nature of this interaction in classical models. The optimized bond length $\mathrm{CO-H_2O}$ (Figure 4a) for the $A_I$-mode is $d_{\mathrm{CO-H_2O}} = 1.8\,\mathrm{\mathring{A}}$ and it coincides if calculated with TB and DFT-GGA. The present work emphasizes the importance of an atomistic model of the solid–liquid interface that simultaneously reproduces the interfacial water structure/reactivity and the underlying quantum nature of semiconductor materials with electronic correlation effects such as polarization and charge transfer.

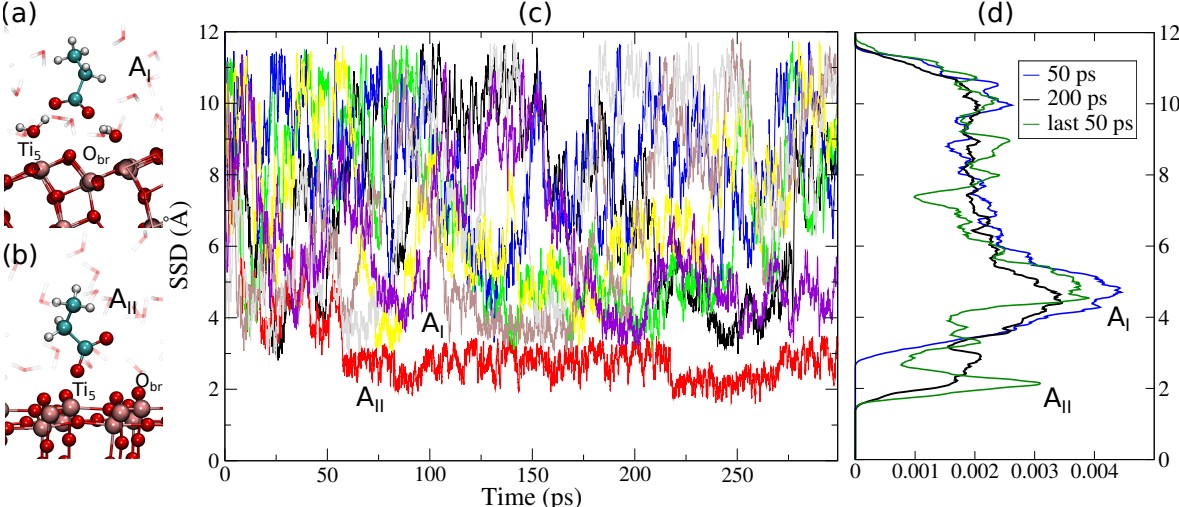

**Figure 4.** (**a**,**b**) Snapshots of adsorption modes $A_I$ at SSD = 4.5 Å and $A_{II}$ at SSD = 2 Å for Aspartate on anatase (101). The freely diffusive water layers begin at SSD = 6 Å. (**c**) The surface separation distance (SSD) for the eight walkers used in the free energy calculations of aspartate. The SSD is sufficiently sampled to distinguish adsorption mode $A_I$ after 40 ps. (**d**) The cumulative SSD distribution from all walkers at different times shows that the system is still not converged after all walkers have been simulated for 300 ps.

The present work has showed that the force estimator, Equation (3), is superior to the traditional Equation (2) for calculating PMFs with metadynamics. Although advantages of the force estimator have been discussed previously [37], these becomes critically important in quantum–chemical simulations with limited sampling time. The slow convergence of the PMFs with the accumulated bias potential as an estimator of the free energy is due to that AWTMetaD is diffusion-limited. Standard thermodynamic integration (including umbrella sampling), on the other hand, is inefficient when the important regions are unknown prior to the simulation, as is usually the case. MetaDF improves sampling by combining these techniques so that the bias potentials provide nearly uniform sampling along the collective variable and create an optimal configuration set for the thermodynamic integration. The multiple walkers boost the sampling of the relevant regions of the CV thus representing the optimal tool for the force estimator. As well-tempered MetaD can suffer from poor sampling if the Gaussian heights became too low before the bias potential reaches the optimal shape, using fixed bias for the metadynamics could eventually speed up even more the convergence. Furthermore, the key point of MetaDF is to collect the forces in the whole relevant region of the free energy landscape, but not to provide the bias potential which exactly compensates the free energy profile. The augmenting collective variable (orientation of the adsorbate molecule) and the multiple walkers further improve

sampling of orthogonal ("slow" or "hidden") degrees of freedom. The mean force as a function of the CV is not affected by the bias, so the forces in a small region (bin) of the CV can be averaged in time. Gaussian bias potentials are inserted every 25 fs, but forces are collected at each time step. Our case study estimates that once the bias potential has explored all values of the CV in the range of interest, force integration converges at least 100 times faster than AWTMetaD with the standard free energy estimator. The MetaDF method can therefore be used in all situations where sampling is limited by diffusion or by strongly bound states.

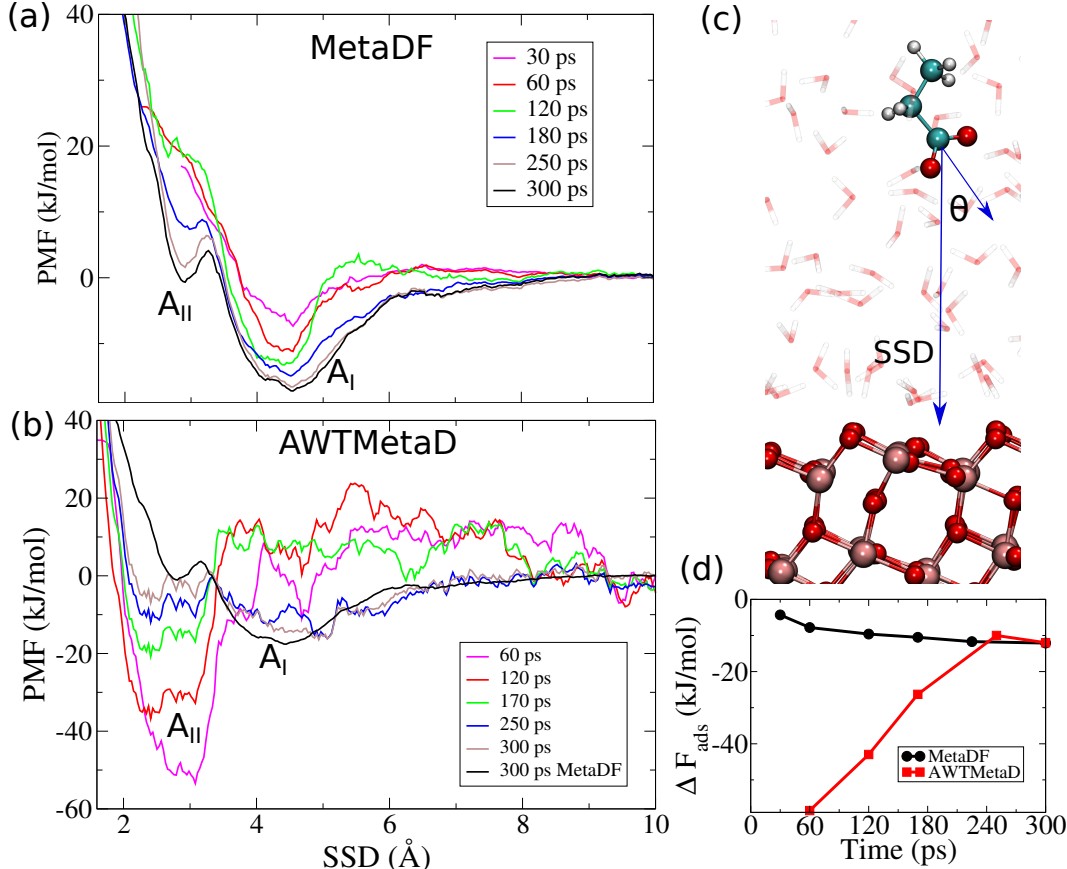

**Figure 5.** Potentials of mean force (PMF) along the surface separation distance (SSD)for Aspartate on anatase (101) calculated with (**a**) MetaDF and (**b**) AWTMetaD. The two minima correspond to adsorption modes $A_I$ (indirect surface contact) and $A_{II}$ (direct surface contact). Adsorption mode $A_I$ converges within an error of $3.5\,\mathrm{kJ\,mol^{-1}}$ in 180 ps with MetaDF (as estimated from the force variance [57] and the convergence profile (**d**)), but fails to converge after 300 ps with AWTMetaD. (**c**) A representation of the two CVs used to sample the free energy landscape. $\theta$ is the angle between the normal vector to the TiO$_2$ surface and the vector $C_{OO^-} - C_{fn}$. (**d**) $\Delta F_{ads}$ as function of time. MetaDF converges asymptotically after 180 ps but AWTMetaD has still not converged after 300 ps.

## 4. Conclusions

There is a pressing need for simulation protocols that target strongly adsorbing molecules, e.g., amino acids on TiO$_2$ surfaces, so that adsorption free energies can be accurately determined for such situations. MetaDF is a sampling method that combines metadynamics and thermodynamic integration to accelerate the convergence of PMF calculations. Multiple walkers and augmented collective variables further improves sampling to a point where free energies can be determined with high accuracy, even in cases with strong adsorption. We tested two Tight-Binding parameterizations for adsorption of the charged Lysine and Aspartate amino acids on the anatase (101) surface. For both molecules, we found large adsorption free energies compared to previous CMD studies.

We hypothesize that the quantum nature of these systems strongly influence the adsorption behavior due to polarization and charge transfer at the interface. For the present case, the PMFs converge within 180 to 250 ps (per walker) in MetaDF simulations with eight walkers. This time scale is accessible to large-scale ab initio molecular dynamics simulations with GGA-level density functional theory, which opens the possibility to move beyond the simplifications of tight-binding DFT and calculate adsorption PMFs at bio-inorganic interfaces with full electronic treatment. MetaDF is particularly useful in situations with limited sampling, such as ab initio simulations, or when diffusion is hindered, e.g., by bound states or barriers along hidden degrees of freedom. Hopefully, MetaDF will be extensively used incoming systematic investigations of various kinds of bionano interfaces.

**Supplementary Materials:** The following are available online at www.mdpi.com/xxx/s1. 2D free energy maps (Figure S1) and the variation of Gaussian insertion as function of time (Figure S2) can be found in Supporting Information.

**Author Contributions:** All the authors contributed to the conceptualization and the development of the methodology. L.A. computed all the calculations, the formal analysis and the validation; further he produced all the figures and drafted the first original version of the manuscript. E.G.B. and A.L. revised all the data and the analysis, contributed to write the final version of the manuscript and provided a constant supervision to the project. All authors have read and agreed to the published version of the manuscript.

**Funding:** The authors received no external funding for the research, authorship, or publication of this article.

**Acknowledgments:** This study has been supported by the Swedish Research Council (Vetenskapsrådet), grant 2017-03950, and by the Horizon2020 program (SmartNanoTox project). The computations were performed on resources provided by the PRACE 15-th project call allocation (BioTitan project), and by the Swedish National Infrastructure for Computing (SNIC), through the Center for Parallel Computing (PDC).

**Conflicts of Interest:** The authors declare no conflict of interest.

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
