# Peer review of "Improved Sampling in Ab Initio Free Energy Calculations of Biomolecules at Solid–Liquid Interfaces: Tight-Binding Assessment of Charged Amino Acids on TiO2 Anatase (101)"

_computation, doi:10.3390/computation8010012_

Round 1

Reviewer 1 Report

The authors proposed a new sampling method called MetaDF to accelerate the PMF convergence. The new method has been properly validated and could be applied to calculate adsorption free energies with ab initio MD simulations hopefully. I just have a few concerns that may be addressed by the authors:
(1) The unphysical deprotonations reported (paragraph 1, page 3) indicate the Matsci parameterization cannot well describe the Lysine and Aspartate adsorptions at TiO2/water interfaces. In principle, the Matsci parameters should be reparametrized. However, the authors added harmonic constraints to avoid the unphysical deprotonations. Not sure if the Matsci parameterization with harmonic constraints can give correct adsorption free energies. If not, the conclusions may change.
(2) The authors say "For the DFT calculations we used the same set-up of parameters implemented in our previous publications[32]". For the convenience of readers, the computational details of DFT calculations should be given.
(3) The authors also say "As augmenting variables, we used the angles...". The authors should discuss how the angles were chosen as augmenting variables.

Reviewer 2 Report

-This is a slightly edited version of the review I previously submitted to a different journal-

The manuscript by Agosta et al describes the application of a newly developed variant of metadynamics, which is described as giving a faster convergence of the method, particularly for DFT calculations. This methodology was then applied to the adsorption of small organic molecules on anatase using a Tight Binding approach.
Although the main scientific conclusion that TB simulations indicate a much stronger adsorption of the organic molecules on anatase is probably valid, I don't think there is enough evidence that the new proposed method speeds up significantly the convergence of the free energy calculation.
I find the discussion of calculations confusing and often misleading, which overshadows any scientific results presented here.
I would therefore recommend the authors to rewrite the parts of the paper to focus more on the scientific part, and reduce the emphasis on the method, which appears as fast converging as the standard well-tempered multiple-walkers methadynamics. Discussing any charge transfer between the surface and the adsorbateobserved during the DFT optimisation would also make the paper more relevant.

In the revised version of the manuscript the authors added a test of the convergence of the new proposed method (MetaDF) compared with standard well-tempered metadynamics (AWTMetaD), figure 1, which clearly shows that the two methods converge to a statistically equivalent free energy after 200ns. However, when comparing the free energy profiles computed after 1, 3 or 30ns with the two methods there is also no significant difference in the convergence rate. At 30ns the free energy curves have diverged a bit at short range, but they appear at similar distances from the converged curve, indicating that there is little difference in the convergence rate.

The 300ps MetaDF with 8 walkers appears to converge faster, but here I think the discussion is misleading, because each walker has run for 300ps, hence the total simulation time is 2.4ns. Therefore, the convergence of this calculation has to be compare with something in between the 1 and 3 ns run. Unfortunately the authors did not show an analogous multiple walkers calculation for the standard well tempered metadynamics, but I would imagine the results would be very similar. In fact any apparent improvement to the convergency of the free energy calculation is purely due to the use of the multiple walkers. This calculation is performed by running 8 parallel simulations starting from different parts of the free energy surface, therefore even if each walker diffuses only a short distance, most of the free energy landscape is somewhat covered, and this has, in my opinion, nothing to do with the MetaDF method developed by the authors.

A flat profile is not a signature of the convergence of the entire free energy surface, as stated on page 11. It just indicates that the flat region is probably homogeneously converged but it doesn't mean anything for the relative free energy of the L1 and L2 states or any free energy barrier that separates them. This is clearly shown by figure 1 where the right had side of the free energy remains flat regardless of the simulation time, while the height of the Mdir state relative the desorbed state constantly changes.

In many parts of the paper the authors mention DFT-GGA, which I find misleading. There is no ab initio free energy calculation performed in this manuscript, and the only ab initio calculations are a few geometry optimisations. Also the title mention ab initio free energy calculations, which is deceptive, Tight Binding is different from DFT. Also a free energy calculation based on 8 walkers running for 300ps each, is stil beyond what can be achieved in a reasonable time by DFT.

On page 12, the authors estimated an error on the free energy of about 3kJ/mol based on a theory which seems to have been developed for Umbrella Sampling simulation. Is there a proof that this works also for multiple walkers metadynamics?

Is the MetaDF also run with adaptive gaussian widths?

Figure 4, as indicated by the authors, show that 300ps times 8 walkers is not enough to guarantee that the free energy calculation is converged. Hence, it is a big leap of faith to assume that the other calculations are. The fact that a free energy profile is smooth in the other cases is not really a guarantee that the relative height of all the portions of the free energy is converged.

The authors should also explain why they have chosen to use the metadynamics with adaptive gaussians' widths, which is not really standard. Does it provide any improvement with respect to the normal fixed-width Gaussians?

Round 2

Reviewer 1 Report

The authors have addressed the reviewers' concerns and thus the manuscript can be published as is.